# MULTIPLE ENCODER-DECODERS NET FOR LANE DETECTION

## ABSTRACT

For semantic image segmentation and lane detection, nets with a single spatial pyramid structure or encoder-decoder structure are usually exploited. Convolutional neural networks (CNNs) show great results on both high-level and low-level features representations, however, the capability has not been fully embodied for lane detection task. In especial, it's still a challenge for model-based lane detection to combine the multi-scale context with a pixel-level accuracy because of the weak visual appearance and strong prior information. In this paper, we we propose an novel network for lane detection, the three main contributions are as follows. First, we employ multiple encoder-decoders module in end-to-end ways and show the promising results for lane detection. Second, we analysis different configurations of multiple encoder-decoders nets. Third, we make our attempts to rethink the evaluation methods of lane detection for the limitation of the popular methods based on IoU.

## 1 INTRODUCTION

Autonomous driving has become a focus of computer vision nowadays. A key component of autonomous driving, optical image based lane detection, is also one of the most important task for feature maps producing and Semantic information utilizing research. In practice, camera-based lane detection still be a challenging task for several reasons. First, the local information of a lane such as sharp, edges, texture and color, can not provides distinctive features for lane detection. Second, it is common for vehicles to be in the complex optical scenarios like rainy, night, shadow, dazzle light conditions. Third, the incomplete lanes make it difficult to get the local informations while in a condition such as crossroad, crowded traffic, no line, etc. Furthermore, the strong global prior knowledge of lane detection put forward higher requirements for the perceptive field of CNNs.

End-to-end CNNs always give better results than systems relying on hand-crafted features. Although the low-level feature maps can supply the local information of the whole image, hand-crafted features based systems are limited to generate sharp features. Most existing algorithms for the segmentation of road boundaries and lane detection (e.g. (Borkar et al., 2012), (Deusch et al., 2012), (Hur et al., 2013),(Jung et al., 2013), (Tan et al., 2014)) rely on detectable and meaningful parts of the local features. As aforementioned limitations, it insufficient to recognize and detect lanes using local features in lots of scenes. Highly hand-craft features based methods can only deal with harsh scenarios.

CNNs show the robust capacity to capture object localization on image classification (Krizhevsky et al., 2012),(Sermanet et al., 2013), (Simonyan & Zisserman, 2014) and object detection (Girshick et al., 2014),(Liu et al., 2015), (Redmon et al., 2016), (He et al., 2016), but less explored on Semantic Image Segmentation due to strong prior information is needed. Although High-level features have shown great capacity to understand Semantics, they lost localization accuracy of the object. Especially in lane detection task, while a human being is able to recognize the lane well under different scenes, could be very challenging for the weak visual appearance and strong prior information conditions, including crowded traffic, dim or dazzle light, etc.

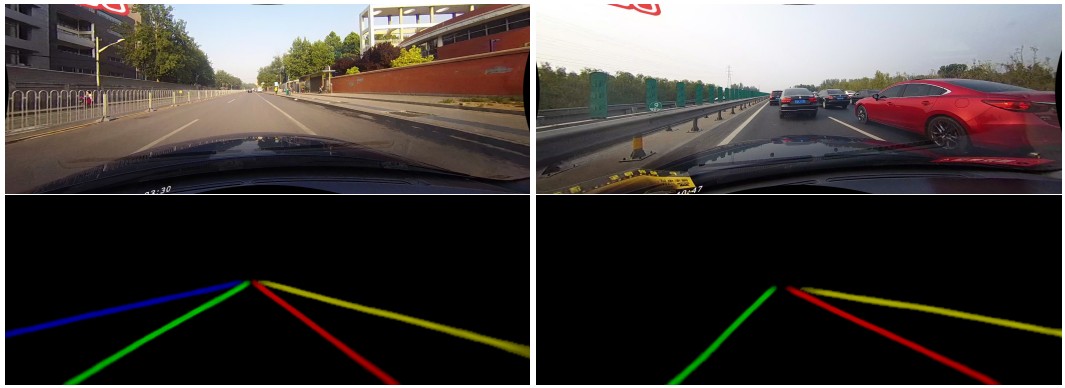

Figure 1: Example results of our method on the task of lane detection.

## 1.1 RELATED WORK

For lane detection, traditional methods rely on highly-specialized, handcrafted features and heuristics from the background pixel by pixel to identify lane segments. These works detect generic sharps of markings and try to fit a line or a spline to localize lanes by Kalman filter(Nieto et al., 2011), tracking-based method(Lipski et al., 2008), B-snakes(Sawano & Okada, 2006),(Wang et al., 2004), hierarchical bi-partite graph(Nieto et al., 2008), color-based features (Chiu & Lin, 2005), the structure tensor (Loose et al., 2009), the bar filter (Teng et al., 2010), ridge features (Lpez et al., 2010), etc. In general, these traditional approaches are prone to robustness issues due to road scene variations that can not be easily modeled by such model-based systems.

Popular choices of recent methods have replaced the feature-based methods with model-based methods. In an attempt to take advantage of the CNN features from multiple layers, (Huval et al., 2015) used object detection methods to detect lanes, (Pan et al., 2017) views rows or columns of feature maps as layers and applies convolution, (Neven et al., 2018) try to cast the problem as an instance segmentation problem,(Ghafoorian et al., 2018) applied a GAN framework to mitigate the discussed problem. (Cong et al., 2018) tackle this problem by using template matching with RGB-D camera.

There have been some other attempts to parse semantic information in a pixel level. (Visin et al., 2015) and (Bell et al., 2016) utilized recurrent neural networks(RNN) to pass information along each row or column, thus in one RNN layer each pixel position could only receive information from the same row or column. (Liang et al., 2016) proposed variants of LSTM to exploit contextual information in semantic object parsing, but such models are computationally expensive. Researchers also attempted to combine CNN with graphical models like MRF or CRF, in which message pass is realized by convolution with large kernels (Peng et al., 2017).

## 1.2 OUR CONTRIBUTIONS

Our contributions can be summarized to the following.

First, reduced localization accuracy due to the weak performance of combining the local information and global information effectively and efficiently. We propose an end-to-end net architecture which is applied to multiple encoder-decoders module. Multiple encoder-decoders net shows a successful result for lane detection.

Second, we analysis the capability of nets with different configurations to combine the location and semantic information. In both visual and statistical way, we compare the performances of nets with different encoder-decoders times and give an explanation of the performances.

Third, the evaluation methods which are used in recent works for lane detection are transfered from other tasks. there exists some limitations of these methods for lane detection. We make our attempts to rethink these IoU based methods.

## 2 MULTIPLE ENCODER-DECODERS NET

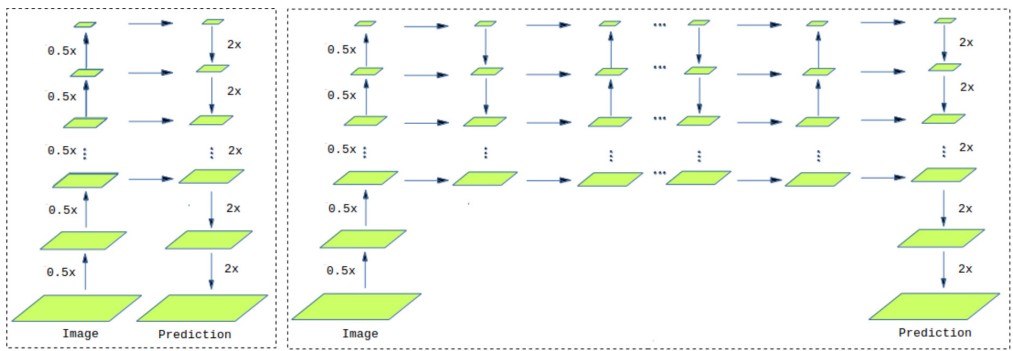

(a) Single encoder-decoder net      (b) Multiple encoder-decoders net

Figure 2: (a)Nets with multiple encoder-decoders model can combine richer local and global information than nets with a single encoder-decoder model.

Recent methods to do the semantic image segmentation task are based on one single spatial pyramid model or encoder-decoder model. Encoder layers are used to get the semantic information and the decoder layers are utilized to get the localization information. Some current state-of-the-art approaches rely on CRF/MRF methods to improve localization performance or enlarge the kernel to increase the object sharp accuracy. But in fact, the low-level feature maps are able to supply the local information of the whole image and the high-level feature maps can offer the sharp information fully, which we used multiple times to extract the semantic information and pinpoint the localization of the objects.

CNNs have achieved great success on producing feature maps and utilizing semantic information. For lane detection task, the challenge is to combine the local information and global information effectively and efficiently. We address the issue by merging the feature maps multiple times as shown in figure 2.

The traditional methods are limited in the little receptive field of the small kernel and the huge amount of computation of the large kernel. But for multiple encoder-decoders nets, it is possible to use the small kernel for low-level features and the large kernel for high-level features. In practice, we use the 9*9 kernel for the lowest-level features for only once and the 3*3 or 1*1 kernel for the most feature maps, which is able to learn the spatial and semantic information well and reduce the computation greatly.

To merge the feature maps sufficiently, we not only apply multiple encoder-decoders net, but also place the residual layer and route layer instead of pooling layer and activity layer. It can be seen the combination of the different level feature comprises richer information, might be a better structure to model the lane representation.

During training we optimize the following loss function

$$loss = \frac{1}{N} \sum_{i=1}^{N} \sum_{j=1}^{S} (\hat{x}_{i,j} - x_{i,j})^2$$

$N$ denotes the number of lanes of the input image, while $S$ denotes the number of pixel. $x_{i,j} = 0.5$ if a lane covers the pixel of $i, j$, and if a lane does not cover the pixel of $i, j$ $x_{i,j}$ is set to 0. Considering the imbalanced label between background and lane markings, the loss of background is multiplied by 0.5.

## 3 EXPERIMENT

In this section, we compare the influence of the different level encoder-decoder times and number of channels of a multiple encoder-decoders net on different scenarios. Furthermore, we give the vi-

sualized results of comparison between our method and the state-of-the-art methods in figure 3. We modify the figure from (Pan et al., 2017) directly so that the results is able to prove the significance of multiple encoder-decoders net.

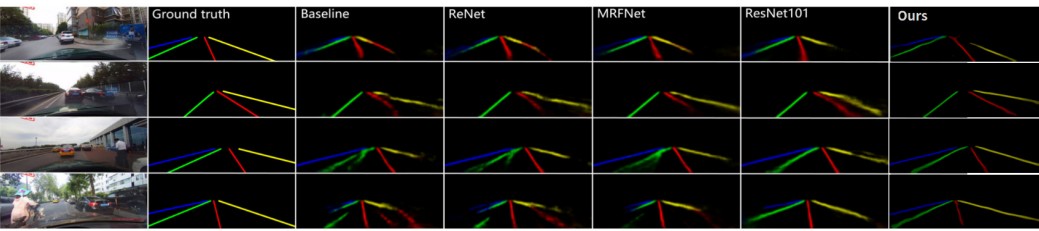

Figure 3: The visual results of comparison between our method and the state-of-the-art methods, which can tell the nets with multiple encoder-decoders model can combine richer local and global information than nets with a single encoder-decoder model.

## 3.1 EVALUATION

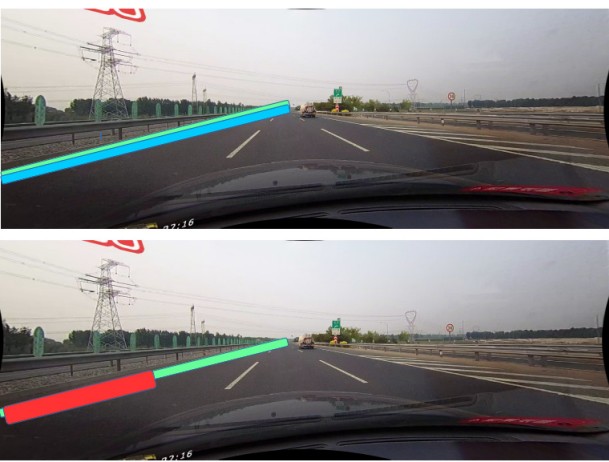

Figure 4: IoU based methods. The green line denotes ground truth, while the blue and red lines both denote lines with IoU equal to 0.33

Recent works evaluated lane detection methods based on IoU, calculating the intersection-over-union (IoU) between the ground truth and the prediction. Predictions whose IoU are larger than certain thresh- old are viewed as true positives (TP). Evaluation methods based on IoU such as Precision, Recall and F-measure as follows have limitations to evaluate the predictions.

$$Precision = \frac{TP}{TP + FP}$$

$$Recall = \frac{TP}{TP + FN}$$

$$F - measure = (1 + \beta^2)\frac{Precision Recall}{\beta^2 Precision + Recall}$$

As figure 4 shown, predictions with the same IoU can be different extremely. while the red line is inaccurate, the blue line is an apparent better prediction. We have thought about adding geometrical constraint, evaluating by key points and some other probability theory methods, but these methods are less than ideal either. In order to judge whether a lane marking is successfully detected and discover the the ability to learn the correct and precise features of different nets, we have compared

more than 500 probmaps of each level nets manually and count the accuracy of these probmaps as shown in figure 7.

## 3.2 DATASET

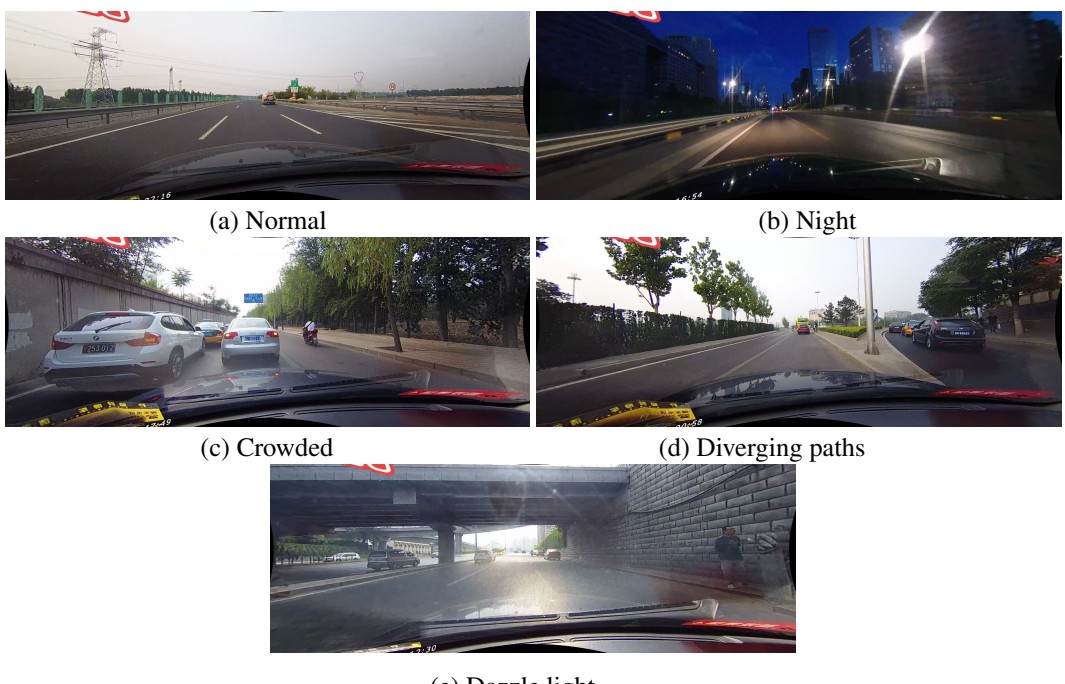

(a) Normal (b) Night

(c) Crowded (d) Diverging paths

(e) Dazzle light

Figure 5: (a)These samples are sampling randomly from the CUlane dataset of normal, night, crowded , diverging paths and dazzle light scenarios.

We verify our method on the CUlane (Pan et al., 2017), the current biggest dataset of lane detection, which has more than one hundred thousand images. figure 5 shows samples of different conditions of the dataset including normal, night, crowded , diverging paths, dazzle light, etc . And the corresponding predictions are given in figure 6 and figure 7. These samples are sampling randomly from the dataset of the respective scenarios.

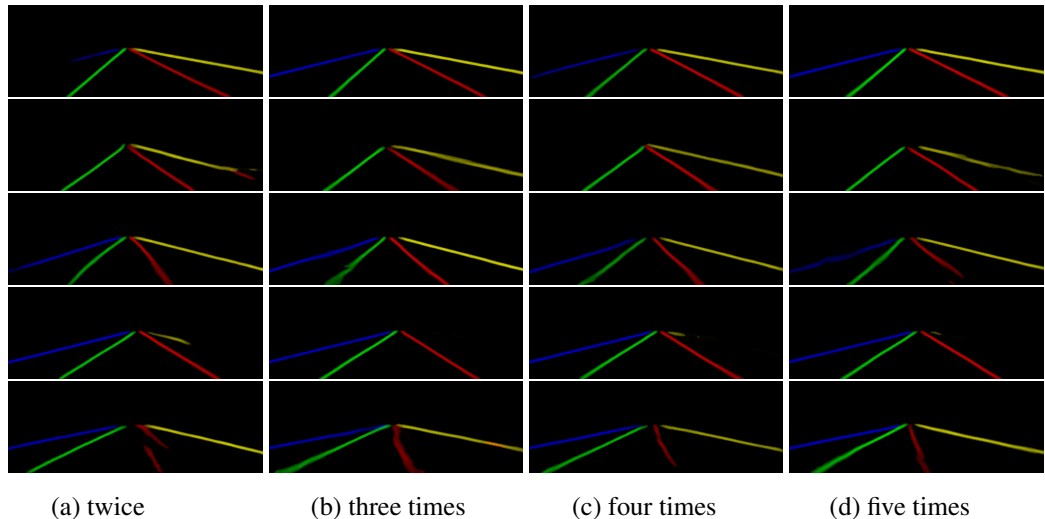

(a) twice (b) three times (c) four times (d) five times

Figure 6: (a)lane detection promaps of multiple encoder-decoders nets with different configurations.

### 3.3 NUMBER OF ENCODER-DECODER

After abundant experiments, generally, we find that ability of the nets to acquire semantic information enhances along with the number of encoder-decoder growing, but the ability to extract the spatial information and pinpoint the localization of the objects declines. Nets with three or four times encoder-decoder model reach a balance between the local and global information which can be seen in figure 6 and figure 7.

For the nets we used , the twice encoder-decoders model has 56 layers, three times encoder-decoders model has 79 layers, four times encoder-decoders model has 102 layers and five times encoder-decoders model has 125 layers. As what can be seen in the the column chart, for normal scenarios, these nets achieve competitive result on predictions. On the contrary, nets with three or four times encoder-decoder model get much better results for complex optical conditions. On the other hands, the very deep structure of five times encoder-decoders model plays a positive role of modeling the global feature for diverging paths.

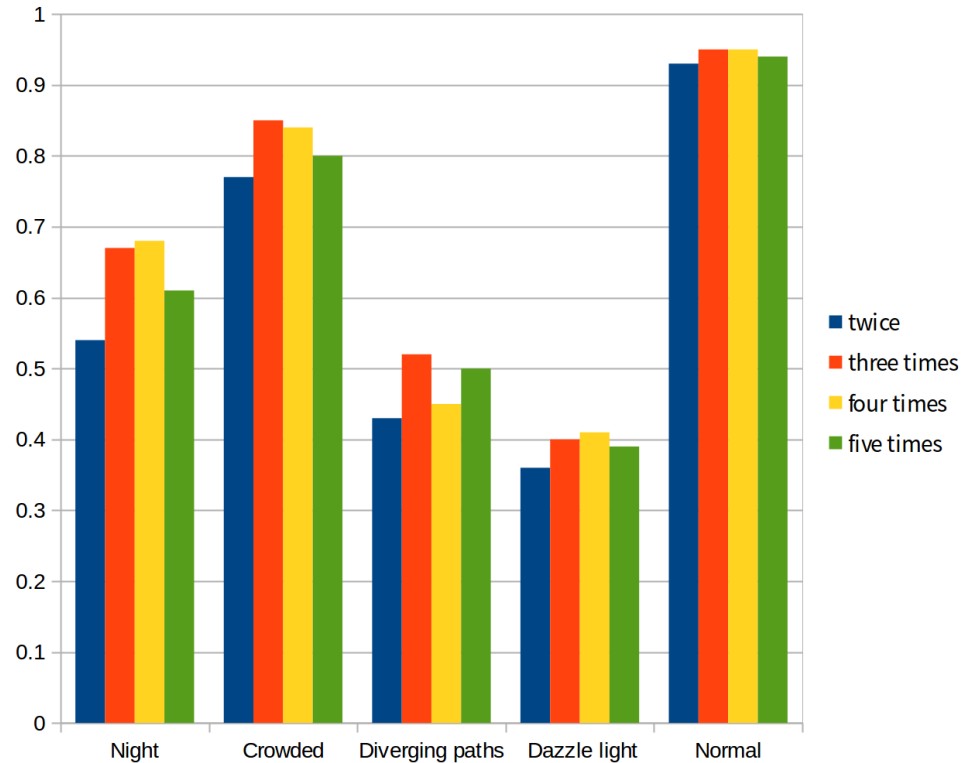

Figure 7: lane detection accuracy of multiple encoder-decoders nets with different configurations.

### 3.4 CHANNEL OF LAYERS

For popular networks (Simonyan & Zisserman, 2014) (He et al., 2016), a large quantity of channel is stacked in network architecture design. A general way is that channel increases gradually along with the depth of the nets increase s in the first few layers, and a large quantity of channel (eg., 256, 512 or 1024) is used in the rest of layers.

To improve ability of the network, we propose a small quantity of channel to reduce overfitting by considering interdependencies among channels. We enable networks with tens of layers to train easier and get a better performance by stacking a small quantity of channel(eg., 4) more times in the entire network.

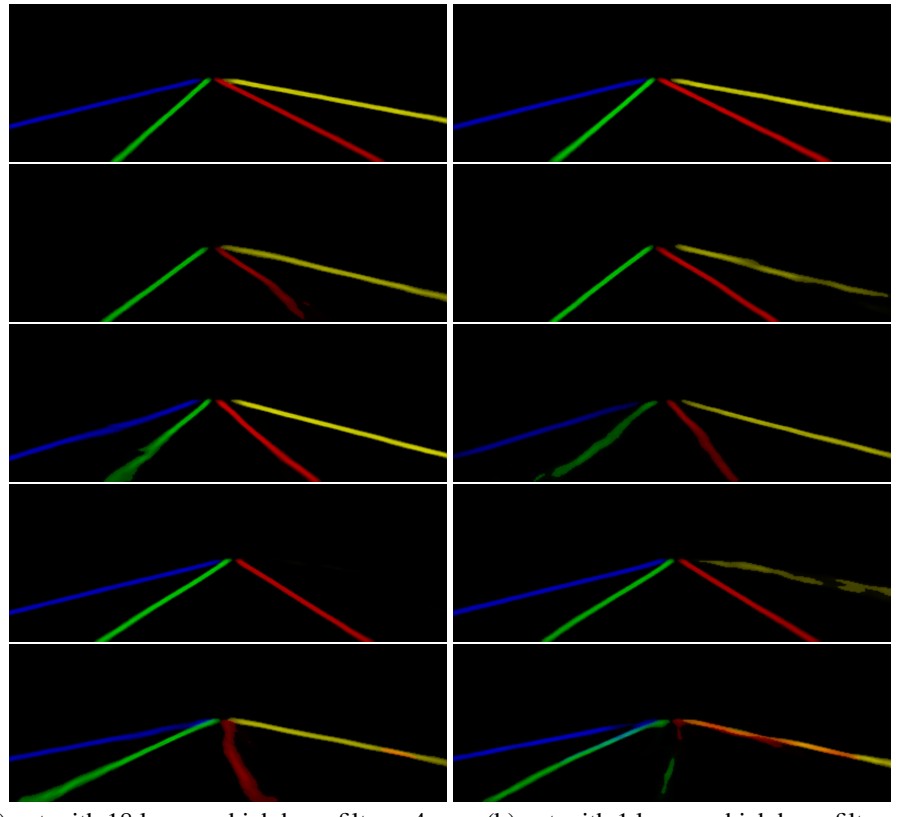

(a) net with 18 layers which have filters=4          (b) net with 1 layers which have filters=4

Figure 8: Example results of our method with three times encoder-decoder model and different number of channel.

## 4    CONCLUSIONS

We propose multiple encoder-decoders net to extract the spatial information and pinpoint the localization of the lane detection. multiple encoder-decoders structure allows the net to improve localization and sharp performance simultaneously. Furthermore, to improve ability of the network, we propose a small quantity of channel to reduce overfitting by considering interdependencies among channels. Extensive experiments on CUlane dataset demonstrate the effectiveness of our proposed net. Last, we rethink the evaluation methods which are used in recent works and exist some shortcomings for lane detection.

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
