# OpenReview forum: "Multiple Encoder-Decoders Net for Lane Detection"
_ICLR.cc/2019/Conference_

### Official Review · AnonReviewer3 · 2018-11-01
**Interesting idea should be explained better, and lacks objective comparison to baselines**

**Rating:** 4
**Confidence:** 4

**Review:**

The paper addresses the problem of pixel-wise segmentation of lanes from images taken from a vehicle-mounted camera. The proposed method uses multiple passes through encoders decoders convnets, thereby allowing extract global features to inform better local features, and vice versa. Only qualitative baseline comparisons are presented by manually comparing the output of the network to reported results of other methods in [Pan et al.2017].
It is unclear to me if the proposed multiple encoder-decoder network is a novel architecture, or a known architecture applied to a novel use case. In case of the former, more details should be given on the design of the network, how it is trained, etc. for reproducibility. The biggest problem however is the subjective manual comparison to existing methods, which the authors do in favor of a quantitative comparison using well-understand objective metrics. While they point out problems with evaluating segmentation with conventional accuracy metrics, no attempt is made to make a better objective measure. We are left to judge the results on only a few selected example frames.
It is also unclear how the method and evaluation strategy compares to methods which predict lanes as splines or other parameterized functions. E.g. see surveys on existing approaches, and discussion of different evaluation strategies, e.g. "Recent progress in road and lane detection: a survey" [Hillel et al.2014] and "Visual lane analysis and higher-order tasks: a concise review" [Shin, 2014].
Throughout the paper, various fuzzy and unclear statements are made (see detailed comments below). The paper would be in a better shape if more time is spend to improve the writing, provide more details on the method, and extend the experiments.

Pros:
+ multiple encoder-decoder stages could be beneficial for lane segmentation

Cons:
- lacking evaluation and comparison to baseline methods
- missing details on proposed network architecture, making it hard to reproduce
- unclear what colors in figures for qualitative evaluation represent: are individual lanes also distinguished?

Below are more detailed comments and questions:
* Abstract
	* "the capability has not been fully embodied for" → Fuzzy statement, I don't understand what this means.
	* "In especial" → check grammar
* Sec 1.: Introduction
	* "the local information of a lane such as sharp, edges, texture and color, can not provides distinctive features for lane detection" local edges are not distinctive for lanes? Possibly local edges alone are not sufficient, but various lanes detection approaches rely on edge extraction as features. This statement therefore seems too strong.
	* "End-to-end CNNs always give better results than systems relying on hand-crafted features.". It is not possible to say that one type of classifier categorically better than another. The 'best' classifier depends on the problem at hand, valid assumptions that can be made, and the amount of training data avaiable, among others. For instance, "How Far are We from Solving Pedestrian Detection?" [Zhang,CVPR16] demonstrates that CNNs do not always give better results than hand-crafted features for some tasks and datasets. The paper should be more careful with such strong statements.
	* "Highly hand-craft features based methods can only deal with harsh scenarios.". I don't understand, is this statement intended as an argument against hand-crafted features? Isn't it good to deal especially with harsh scenarios?
	* "but less explored on Semantic Image Segmentation due to strong prior information is needed." CNNs are extensively used for semantic image segmentation, e.g. see the well-known Cityscapes benchmark.
	* "recent methods have replaced the feature-based methods with model-based methods.". Not sure why the paper call CNNs "model-driven methods", but refer to the earlier classical methods with highly designed representations (Kalman filter, B-snakes, ...) as "feature-based methods". This seems diferent from what I typically see, where CNNs are referred to as 'data-driven methods', and the classical methods as "model-driven".
* Sec 1.2: Contributions
	* "First, reduced localization accuracy due to the weak performance of combining the local information and global information effectively and efficiently". Instead of presenting a first contribution, the paper presents a problem. Do the authors mean that they "tackle the problem of reduced localization accuracy ..." ? That would still not make this contribution very concrete though ...
	* "We make our attempts to rethink these IoU based methods." → Please argue in favor of your new method. An in-depth comparison of evaluation methods, and why some metrics fail or could be redesigned would be good. However, the paper currently fails to present a new metric, and convince that it tackles shortcomings of established metrics.

* Sec 2.: Multiple Encoder-Decoder Nets
	* Figure 2: Is this the first paper to propose this multiple encoder-decoder net? Or is the idea taken from other work, and is the novelty to apply it to this problem? If this general architecture was already proposed (for semantic segmentation?), please add citations and discuss it as related work. If this network design is completely novel, I would expect more details on how the network is constructed (e.g. dimensions of each layer, non-linear activation function used, batch normalization, strides, etc.).
	* "the following loss function:". Since it is a binary classification problem, and not a regression problem, why not use a (binary) cross entropy loss instead of a mean squared error?

* Sec 3: Experiments
	* Figure 3: What is the "Baseline" method ? Where are the references to the other works, or is the reader required to read [Pan'2017] to understand your figures?
	* Figure 3: How are the colors in these figures determined? Is this also an instance segmentation problem? From your methodology section I though only binary classification was considered. Do you do some post-processing to separate individual lanes? I find this confusing, as I thought that the task was limited to binary segmentation.
	* "Recent works evaluated ..." please cite the works you refer to.
	* "we have compared more than 500 probmaps of each level nets manually and count the accuracy of these probmaps as shown in figure 7." So if I understand correctly, instead of using an objective evaluation metric, you have reverted to manual labor to visually judge lane detection quality. This is not really a metric, and not really a solution that 'rethinks IoU based methods.' Problems of your approach is that it is unclear on what criteria results are judged, your evaluation is not objectively reproducible by others, and does not scale well for novel future evaluations. Why is this even needed? E.g. why not use some chamfer distance or Gaussian smoothing of the edge map if you want to evaluate near coverage instead of hard boundaries? Or, fit a function through the boundary, and evaluate distance (in meters) to true lane. I find the proper discussion and motivation for manual evaluation over objective metric evaluation lacking.
	* Figure 6: What are the Ground Truth images of each row ? E.g. in the fourth row from the top, should the right-most yellow lane be present or not? As it stands, I can't interpret the columns and see which x times is visually 'better'.
* Sec 3.4:
	* "To improve ability of the network, we propose a small quantity of channel to reduce overfitting by considering inter-dependencies among channels." To improve relative to what? Where are the results comparing large amounts vs small amount of channels? Note that Figure 8 is not referred to in the text, and confusingly compares "18 layers" to "1 layers". Do you mean channels instead of layers? And, how many channels were to obtained the results in the preceding sections?

---

### Official Review · AnonReviewer1 · 2018-11-01
**The topic is interesting, but the submission is still a draft**

**Rating:** 2
**Confidence:** 5

**Review:**


# Summary
This work deals with a computer vision task specific to autonomous vehicles, namely detection of lane markings on the road. The authors propose an encoder-decoder CNN architecture  (typically used for semantic segmentation) for which a parts of the encoder and the encoder are instanced several times (with different weights) in order to better capture the semantic and spacial information from intermediate feature maps. The method is tested on the recent Lane Detection dataset (Pan et al.) and report qualitative results.


# Paper strengths
- the paper deals with a topic of interest for the autonomous driving community
- the authors identify a flaw in the IoU accuracy evaluation metric for lane detection

# Paper weakness
- The paper could be written better. It seems unfinished and some additional proof-reading is necessary to correct the multiple typos across the paper

- There are no quantitative results and no comparisons with baselines and related works, making it difficult to evaluate the performances of this work.

- The authors mention that the current IoU accuracy evaluation metric is flawed on some cases and then propose scanning manually results from 500 prediction maps and report results on that. There is no baseline or other related method considered for this evaluation. Also the original dataset from Pan et al. has ~88k train images, ~9.5k validation images and ~35k test images. It is not clear from which set where the 500 images taken and how representative they are for the entire dataset

- The structure of the proposed architecture is not clear, in particular the merging of the feature maps across multiple encoders. From the diagram in Figure 2 b) it seems that the feature maps are transferred to the neighbour encoder/decoder branch similarly with RNNs. Is this right or only the features maps from the first branch are transferred? Furthermore, the merging is done by element-wise addition or concatenation?
Given the application domain, an important aspect is the computational complexity of the proposed architectures. I would welcome such an analysis in the current work as well.

- I enumerate a few other unclear aspects and improvable points in the paper:
  + why standard encoder-decoder architecture are limited to small receptive field (cf. Section 2), while this approach is not and can use large kernels? Do the authors use kernels of different sizes in the branch? This problem is usually addressed with dilated convolutions or parallel sub-networks of different kernel sizes like in Inception.
  + there is no description of the task and of the meaning of the 4 lane markings.
  + in Section 1 the authors mention that "CNNs show the robust capacity to capture object localization on image classification ... and object detection, but less explored on semantic image segmentation due to strong prior information is needed.". This is not true as semantic segmentation is addressed by means of CNNs since several years already by most of community.
  + why the label of lane pixels is set to 0.5? Such problems are typically addressed with binary-cross entropy and the output of the sigmoid is used as classification score at test time.

# Recommendation
This paper deals with an interesting task, but it's still in a rather draft state. There are several shortcomings of the work that I've mentioned above. I recommend the submission for rejection.

---

### Official Review · AnonReviewer2 · 2018-11-02
**A lack of detail and benchmarking combined with limited accessibility make any contribution difficult to assess in terms of originality and significance.**

**Rating:** 2
**Confidence:** 4

**Review:**

The authors propose a neural network architecture for lane detection in on-road driving. The architecture consists of multiple encoder-decoder stages. This is motivated by a need to overcome limitations of traditional CNNs with respect to considering high-level context while providing pixel-level accuracy. The paper additionally claims contributions in the analysis of the performance of various instantiations of this architecture as well as in considering limitations of the popular IoU metric. Experiments are reported on the publicly available CULane dataset.

The paper addresses a topical problem in autonomous driving and ADAS but is currently let down by a severe lack of accessibility. Much of the evidence corroborating the principal claims of the submission appears to be missing. While a number of CNN architectures do struggle to provide pixel-level segmentation accuracy particularly for objects of certain geometries there exists a whole host of literature regarding attempts to remedy this. The SegNet family of works as well as many works leveraging the now established skip-connection u-net architecture is missing entirely. The related works  section does list a number of recent relevant work but does not succeed in putting this into context given the approach proposed here. This is also not remedied in the experimental evaluation as almost no benchmarking to the established state of the art is performed. The evaluation itself is mainly qualitative and does not serve to convince the reader that the approach offered here is beneficial. For example, how does one choose between the different solutions offered in Fig. 8? When it comes to the quantitative evaluation some important detail appears to be missing. For example, what is a probmap and how is accuracy on these computed to arrive at Fig 7? Much of the experimental detail is also left unclear.

The submission also requires extensive spell and grammar checking. This would significantly improve accessibility, though much remains to be done to make the science case more convincing.

Overall this makes the originality and significance of the work difficult to judge. As it stands I can not recommend publication.

---

### Meta-Review · Area_Chair1 · 2018-12-04
**decision**

**Confidence:** 5
**Recommendation:** Reject

**Metareview:**

As the reviewers point out, the paper is below the acceptance standard of ICLR due to low novelty, unclear presentation, and lack of experimental comparison against the state-of-the-art baselines.